# Prospective assessment of malaria infection in a semi-isolated Amazonian indigenous Yanomami community: Transmission heterogeneity and predominance of submicroscopic infection

**Daniela Rocha Robortella**[1,2], **Anderson Augusto Calvet**[3], **Lara Cotta Amaral**[2], **Raianna Farhat Fantin**[2], **Luiz Felipe Ferreira Guimarães**[2], **Michelle Hallais França Dias**[2], **Cristiana Ferreira Alves de Brito**[2], **Tais Nobrega de Sousa**[2], **Mariza Maia Herzog**[3], **Joseli Oliveira-Ferreira**[3]*, **Luzia Helena Carvalho**[1,2]*

1 Departamento de Parasitologia, Universidade Federal de Minas Gerais (UFMG), Belo Horizonte, Brazil, 2 Instituto René Rachou (FIOCRUZ-MINAS), Belo Horizonte, Brazil, 3 Instituto Oswaldo Cruz, Fundação Oswaldo Cruz (IOC/FIOCRUZ), Rio De Janeiro, Brazil

* luzia.carvalho@fiocruz.br (LHC); lila@ioc.fiocruz.br (JOF)

## Abstract

In the Amazon basin, indigenous forest-dwelling communities typically suffer from a high burden of infectious diseases, including malaria. Difficulties in accessing these isolated ethnic groups, such as the semi-nomadic Yanomami, make official malaria data largely underestimated. In the current study, we longitudinally surveyed microscopic and submicroscopic malaria infection in four Yanomami villages of the Marari community in the northern-most region of the Brazilian Amazon. Malaria parasite species-specific PCR-based detection of ribosomal and non-ribosomal targets showed that approximately 75% to 80% of all malaria infections were submicroscopic, with the ratio of submicroscopic to microscopic infection remaining stable over the 4-month follow-up period. Although the prevalence of malaria infection fluctuated over time, microscopically-detectable parasitemia was only found in children and adolescents, presumably reflecting their higher susceptibility to malaria infection. As well as temporal variation, the prevalence of malaria infection differed significantly between villages (from 1% to 19%), demonstrating a marked heterogeneity at micro-scales. Over the study period, *Plasmodium vivax* was the most commonly detected malaria parasite species, followed by *P. malariae*, and much less frequently *P. falciparum*. Consecutive blood samples from 859 out of the 981 studied Yanomami showed that malaria parasites were detected in only 8% of the previously malaria-positive individuals, with most of them young children (median age 3 yrs). Overall, our results show that molecular tools are more sensitive for the identification of malaria infection among the Yanomami, which is characterized by heterogeneous transmission, a predominance of low-density infections, circulation of multiple malaria parasite species, and a higher susceptibility in young children. Our findings are important for the design and implementation

**Data Availability Statement:** All relevant data are available within the article and its Supporting Information files.

**Funding:** This work was supported by the CNPq-Conselho Nacional de Desenvolvimento Científico e Tecnológico, Coordenação de Aperfeiçoamento de Pessoal de Nível Superior (CAPES) – Finance code 001, PAEF/FIOCRUZ (IOC-FIO-18-2-47 by JOF), Fundação de Amparo à Pesquisa do Estado do Rio de Janeiro - FAPERJ (E-26/110.803/2013 by JOF), Yanomami Special Indigenous Sanitary District (DSEI-Y) for Air transport (by JOF). CFAB, TNS, JOF and LHC are CNPq Research Productivity fellows. Scholarships were sponsored by CAPES (LFFG, LCA and MHFD) and CNPq (DRR and RFF). We also thank the financial support from the Program for Institutional Internationalization of the Higher Education Institutions and Research Institutions of Brazil-CAPES-Print from FIOCRUZ. The funders had no role in study design, data collection and analysis, decision to publish, or preparation of the manuscript.

**Competing interests:** The authors have declared that no competing interests exist.

of the new strategic interventions that will be required for the elimination of malaria from isolated indigenous populations in Latin America.

## Introduction

In the Americas, of the 17 malaria endemic countries, 11 are on target to achieve a ≥40% reduction in case incidence by 2020, while the remaining six are on target for a 20–40% reduction [1]. Despite this overall progress, since 2015, malaria incidence began to rise in the Americas, mainly due to increases in the Amazonian rain forest areas of Venezuela (Bolivarian Republic of), Brazil and Colombia [2]. In 2018, Venezuela alone accounted for 50% of all reported cases of malaria in the Americas, followed by Brazil (23%), Colombia (10%) and Nicaragua (6%) [3]. Malaria control in the Amazon region is difficult due to its low population density and scarcity of reliable transport routes, which make it challenging to deliver and sustain preventive health-care measures [4]. In this context, local control measures need to characterize areas and populations according to their levels of disease transmission, while also considering socio-economic, political and environmental factors.

In the Amazon basin, malaria is endemic among many indigenous people, which is often considered one of the last barriers to malaria elimination due to their geographic isolation [5]. According to the Brazilian Surveillance Secretary of the Ministry of Health (SVS/MS), while malaria increased by 3% among some vulnerable populations living in the Brazilian Amazon, such as gold-miners, in indigenous areas the number of cases increased about 30% over the same period of time (2017–2018) [6]. In this region, indigenous communities, including the traditionally seminomadic and widely dispersed Yanomami ethnic group, suffer from a high burden of infectious diseases [7]. The Yanomami are considered the largest semi-isolated Amazonian indigenous people, which inhabit an area of roughly 74,000 square miles that straddles the Brazilian-Venezuelan border [8]. Partly due to the impact of the current political instability in Venezuela and its associated healthcare crisis, the Yanomami people–who have great vulnerability to malaria, measles, malnutrition and mercury pollution in rivers–are currently confronted with environmental destruction of their homeland and potential ethnocide [9].

In the current study, we prospectively evaluated the prevalence of malaria infection in four Yanomami villages of the Marari community in the northern-most region of the Brazilian Amazon, close to the border with Venezuela. To accurately estimate the burden of malaria infection among the Yanomami, the study design involved the screening of 981 individuals for both microscopic and submicroscopic malaria infection (using species-specific PCR-based assays) during three cross-sectional surveys taken at two-month intervals.

## Individuals and methods

### Study area

The study was performed in a remote forest-dwelling Yanomami community called Marari located in the northern-most region of the Brazilian Amazon in the state of Amazonas (Fig 1). Although only a few studies have so far reported in detail malaria transmission among the Marari community [10], we have previously identified high rates of infection (1.5–2%) in the main local malaria vector, the mosquito *Anopheles darlingi* [8]. Marari is one of the basic Health Units in the Brazilian Yanomami Indigenous Territory, which provide health care for four geographically close communities: Taibrapa-I, Gasolina, Alapusi and Ahima/Castanha, with the last one of these listed located close to the Health Unit (Fig 1). Due to the semi-nomadic habits of the Yanomami, the residents of Taibrapa-I also have an alternative residence named Taibrapa-II. The

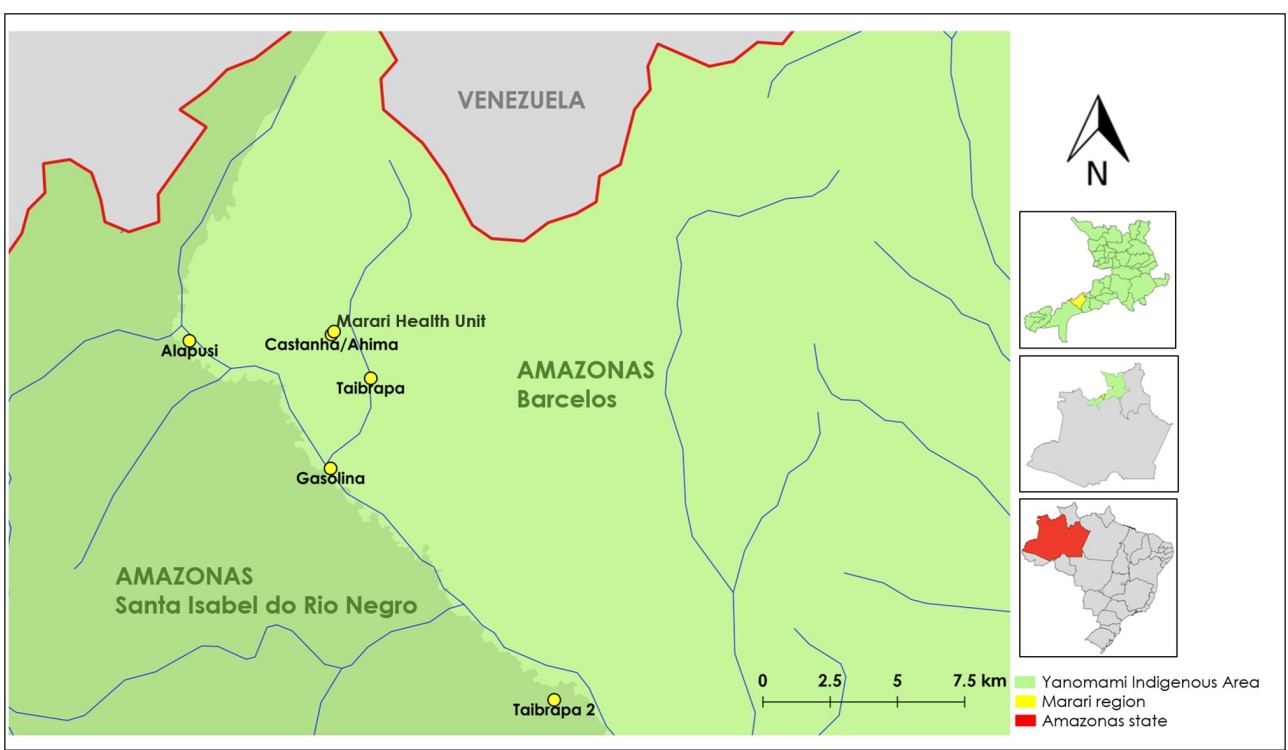

**Fig 1. Map of the Brazilian-Venezuelan border showing the study area.** The Marari community (yellow in the top inset map) is located in the Yanomami indigenous area (green in the inset maps) in the northern-most region of the state of Amazonas (red in the bottom inset map), roughly 454 miles from the state capital, Manaus, in Brazil. The studied villages are marked with yellow dots, while the main rivers are shown in blue. Studied villages shown in the main map are located in the municipality of Barcelos (light green) or Santa Isabel do Rio Negro (dark green). Map sources: the Brazilian Water National Agency, Multi-scale Ottocoded Hydrographic Base, and the Brazilian Institute of Geography and Statistics (IBGE). Amazonas municipal cartographic base (SRC selected- EPSG: 4674, SIRGAS 2000). Scale 1:128.000.

Marari community is located in an area of lowland Amazonian rain forest (139 meters above sea level), which is drained by primary to tertiary tributaries and surrounded nearby by high mountains at the border of Brazil and Venezuela (Fig 1). In Marari, the Yanomami in each village occupy a single large dwelling named a "shabono" or "shapono", which is a circular, thatch-roofed communal shelter with no internal walls [5]. Due to the seasonal dependence on river transport and the high costs of air transportation, access to health services is extremely limited within such Yanomami regions [11, 12]. In our study area, the Yanomami still maintain traditional modes of subsistence, including hunter-gathering and rural agriculture. In 2014, the Annual Parasitological Index (API) for the Yanomami territory was approximately 100 cases per 1,000 inhabitants, with *P. vivax* usually predominating over *P. falciparum* infection [13].

## Study design, participants and sample collection

Ethical and methodological aspects of this study were approved by the Brazilian National Research Ethics Commission (CONEP, Protocol #16907), which regulates studies involving Brazilian indigenous people and was locally supervised by their representatives. The study design involved a longitudinal approach with three cross-sectional surveys at two-month intervals. The first survey (baseline) was carried out in September 2014, the second in November 2014 (2 months later), and the third in January 2015 (4 months after the first baseline survey). At that time of cross-sectional surveys, roughly 30 microscopically confirmed cases per 1,000 inhabitants were officially registered in the study area (Fig 2). Recruitment in the Marari

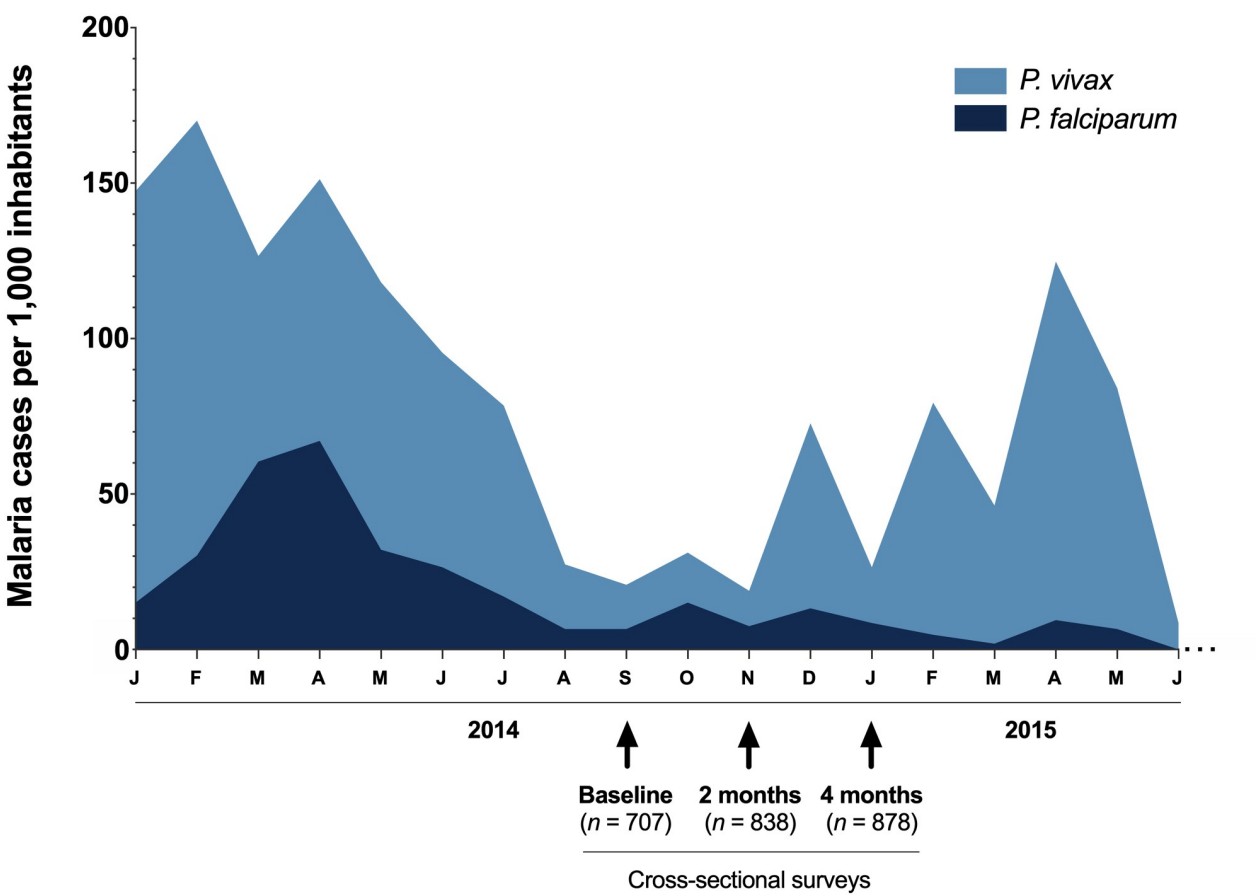

**Fig 2. Monthly-time series of microscopically-confirmed malaria cases in the Marari community of the Yanomami indigenous area (Amazonas, Brazil) during the study period, 2014–2015.** The study design included three cross-sectional surveys: the first survey (baseline) was carried out in September 2014, the second in November 2014 (2 months later) and the third in January 2015 (4 months after the first baseline survey). Data are presented as microscopically-confirmed cases per 1,000 inhabitants, and were provided by the National Malaria Surveillance System Registry (SIVEP-Malaria) from the Brazilian Ministry of Health. Malaria infection caused by *P. vivax* (light blue) and *P. falciparum* (dark blue) are plotted by month.

community involved a bilingual interpreter that explained to the leaders and/or indigenous representatives, the purpose of the study, the procedures to be carried out, and requested written informed consent of each adult participant, or from the next of kin or guardians on the behalf of minors. Blood for malaria parasite detection was collected by finger prick during the surveys. A few drops of blood were placed directly from the finger prick onto either a glass slide for preparation of thick blood smears or Whatman FTA™ classic card (GE Healthcare Life Sciences) for dried blood spots (DBS). The blood spots were dried in open air for 15 min, and stored in zip-locked plastic bags containing desiccant, and then transported to the laboratory at Oswaldo Cruz Foundation (FIOCRUZ) for malaria diagnosis by PCR. At enrollment, age and gender information were recorded for each participant. In total, 981 Yanomami individuals were enrolled in the study, with 707 (72%) recruited in the baseline, 838 (85.4%) during the 2[nd] survey (November 2014) and 878 (89.5%) in the 3[rd] survey (January 2015). During the follow-up period, the minimum and maximum number of participants per village ranged from 197 to 235 (Taibrapa I/II), 174 to 231 (Gasolina), 122 to 154 (Alapusi) and 214 to 282 (Castanha/Ahima). A total of 859 (87.5%) individuals participated in at least two cross-sectional surveys, and 583 (59.4%) subjects were present in all three-cross sectional surveys.

In the current study, malaria infection was defined as the occurrence of any confirmed malaria infection with or without symptoms according to the general guidelines proposed by the WHO Malaria Policy Advisory Committee [14]. Malaria infections were confirmed by either conventional light microscopy (microscopically-confirmed cases) or species-specific PCR-based assays (submicroscopic infections). As malaria symptoms were not evaluated in the study population, the terms "submicroscopic" and "asymptomatic" infection cannot be used interchangeably. Here, submicroscopic infections refer to low density malaria infections below the limit of detection of microscopy, as proposed by the WHO Evidence Review Group on Low-Density Malaria Infections [15]

## Microscopic diagnosis

Thick blood smears were stained with 10% Giemsa solution in phosphate buffer solution at pH 7.2, and examined by light microscopy at the health facilities in Marari by trained microscopists, according to the malaria diagnosis guidelines of the Brazilian Ministry of Health [16]. All microscopically positive cases were treated immediately for malaria, according to the treatment guidelines of the Brazilian Ministry of Health [17].

## Molecular diagnosis

**DNA extraction.** Genomic DNA was extracted from DBS on FTA paper using a commercial extraction kit (QIAamp DNA Blood Mini Kit, Qiagen), according to manufacturer's instructions. DNA was eluted in 150 μl of AE Buffer and stored at −2000B0030C until it was used in the experiments. As an internal control for the DNA extractions, 10% of the extracted samples were submitted to a PCR assay for the amplification of a constitutive human gene (ABO blood group) using primers previously described [18], with modifications for a real-time PCR assay.

**Detection of *Plasmodium* species by PCR-based protocols.** In order to improve the sensitivity of molecular detection of malaria infection, each sample was submitted to at least two PCR-based assays targeting the amplification of different plasmodial genes, i.e., the widely used *18S small subunit ribosomal RNA* (*18S SSU rRNA*) [19] and multi-copy non-ribosomal sequences for the detection of *P. vivax* (*Pvr47*) and *P. falciparum* (*Pfr364*) [20]. The non-ribosomal PCR assays may identify co-infections that could be missed by the ribosomal PCR assay [21]. Detailed PCR-based protocols are included as Supporting Information (S1 File). Briefly, the real-time PCR amplification of non-ribosomal targets was conducted using species-specific TaqMan™ probes and the protocol as we described recently [21]. The amplification of *18S SSU rRNA* was carried-out using the protocol described by Rougemont and colleagues with modification [22], which includes a pair of genus-specific primers with three different internal species-specific TaqMan™ probes, one each for *P. falciparum*, *P. vivax* and *P. malariae*. In case of discordance between the different PCR protocols, samples were submitted to an additional third PCR protocol also based on amplification of *18S SSU rRNA*, as described previously [23]. If no consensus was obtained between the PCR assays, samples were considered as only genus-positive (*Plasmodium* spp.). Concordance between at least two species-specific amplifications was considered the minimum criteria to define mixed-species *Plasmodium* infection.

## Statistical analysis

Demographic, epidemiological and parasitological data were entered into a database created using the Epi Info™ software (Atlanta, GA, USA). For statistical analyses, the following variables were taken into consideration: gender, age, village of origin, malaria infection status at enrollment and during the follow-up. Proportions were compared using 2x2 contingency

tables with either chi-squared tests, adjusted by Yates' continuity correction or Fisher's exact tests, as appropriate. Odds ratios (OR) were used to quantify the strength of the association between variables. The statistical significance threshold was $P < 0.05$, with 95% confidence intervals for all hypothesis tests. These analyses were performed using GraphPad InStat, version 3 (GraphPad Software, San Diego, CA, USA).

## Results

### Malaria infection at enrollment

In our first baseline survey conducted in September 2014, we assessed malarial infection in 707 Yanomami individuals from Marari, with a median age of 14 years (IQR 6–33 years; range 1–79 years) and a male:female ratio of 1:1 (Table 1). Although the number of inhabitants varied between villages (from 122 to 214), subjects were matched by age and gender. The frequency of malaria infection, as detected by optical microscopy, was 1.55% (11 out of 707), ranging from 0.46 to 2.3% between the villages (Table 1). Considering PCR-based protocols, the proportion of positives was higher at 5.95% (42 out of 707), with significant differences between villages that ranged from 1.4% (Castanha/Ahima) to 12.2% (Taibrapa-II) (Table 1). Specifically, the likelihood of having malaria was significantly higher in Taibrapa-II (OR = 9.8, 95% CI 3.05–31.11, $P < 0.0001$) and Gasolina (OR = 5.2, 95% CI 1.64–17.49; $P = 0.0068$) compared with Castanha/Ahima (Fig 3), while the odds of having malaria infection were not significantly different between Alapusi and Castanha/Ahima.

Overall, at enrollment, a similar proportion of *P. vivax* and *P. malariae* infections (35–33%) were found in the study population, followed by *P. falciparum* (16%) and then mixed infections (9%). Of particular interest, *P. malariae* infections were largely restricted to Taibrapa-II village, i.e., 12 out of all 14 *P. malariae* cases detected, suggesting highly localized transmission. Four out of the total of 43 detected infections were mixed: *P. malariae* was present in all 4 of

**Table 1. Baseline demographic, epidemiological and parasitological data from the first survey of the Yanomami villages studied in the Marari community, Amazonas, Brazil, conducted in September 2014.**

| | Characteristics at enrollment | | | | |
| --- | --- | --- | --- | --- | --- |
| | **Taibrapa-II**[*] | **Gasolina** | **Alapusi** | **Castanha/ Ahima**[**] | **Total** |
| | (*n* = 197) | (*n* = 174) | (*n* = 122) | (*n* = 214) | (*n* = 707) |
| **Gender ratio male:female** | 1:1.05 | 1:0.74 | 1:0.79 | 1:1.39 | 1:1 |
| **Age (years)** | | | | | |
| Median (IQR) | 15 (6–35) | 12 (5–30) | 13 (6.25–31) | 15 (8–33.25) | 14 (6–33) |
| Range | 1–76 | 1–69 | 1–76 | 1–79 | 1–79 |
| **Malaria infection, positive (%)** | | | | | |
| Microscopy[***] | 4 (2.0) | 4 (2.3) | 2 (1.6) | 1 (0.46) | 11 (1.55)[§] |
| PCR-based protocols | 24 (12.2) | 11 (6.3) | 4 (3.3) | 3 (1.4) | 42 (5.95)[£] |
| Total | 24 (12.2)[a] | 12 (6.9)[ac] | 4 (3.3)[bc] | 3 (1.4)[b] | 43 (6.1) |

IQR = interquartile range.

Microscopy (thick blood smears).

PCR = polymerase chain reaction.

[*]At the baseline, the inhabitants of Taibrapa were temporarily located in their alternative dwelling (Taibrapa-II).

[**]Marari Health Unit.

[***]No statistical differences (Chi-square test, $P = 0.4614$).

[§-£]Different symbols indicate statistically significant differences between microscopy and PCR (Fisher's exact test, $P < 0.05$).

[a-b-c]Different letters indicate statistically significant differences between the positivity per village (Fisher's exact test, $P < 0.05$).

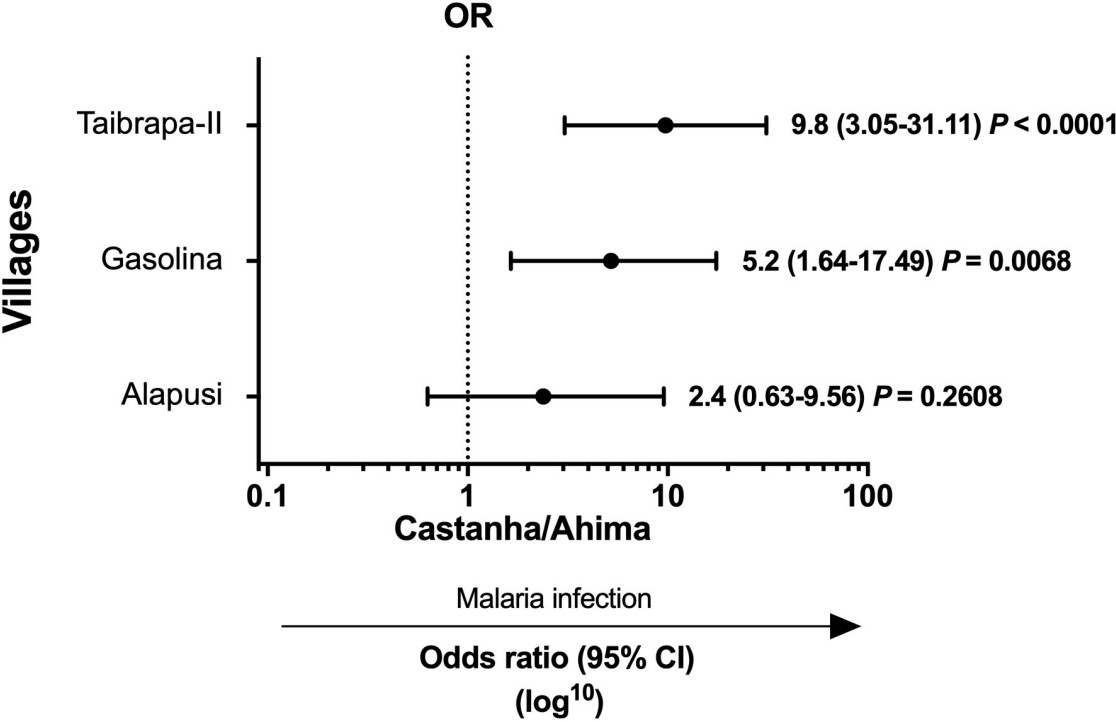

**Fig 3. Odds ratios (ORs) for the relative risk of malaria infection in each village compared with Castanha/Ahima (the village with the lowest number of malaria infections, used as the reference).** At first baseline survey, the inhabitants of Taibrapa were temporarily located in their alternative residence (Taibrapa-II). Crude OR were obtained by using 2x2 contingency tables with a 95% confidence interval (CI). *P* < 0.05 was considered significant. Malaria infection was confirmed by either conventional light microscopy (microscopically-confirmed cases) or species-specific PCR-based assays (submicroscopic or low-density infections).

these mixed infections, while two had *P. vivax* and two *P. falciparum*. In three malaria-positive samples, inconsistencies between different species-specific molecular methods did not allow diagnosis of the particular *Plasmodium* species (i.e., the infections could only be diagnosed as genus-positive).

Only microscopy detected malaria infections in individuals below 16 years old (Fig 4), while PCR-based protocols were able to detect submicroscopic infections in all age groups– including individuals above of 50 years old–showing that submicroscopic malaria infection was prevalent among both children and adults (Fisher's exact test *P* > 0.05 for all comparisons between the different age groups) (Fig 4). The frequency of malaria infection was similar between genders (S1 Fig).

### Spatial and temporal dynamics of malaria prevalence

As the majority of malaria infections among the Yanomami were submicroscopic, we sought to evaluate the spatial and temporal dynamics of malaria infection two and four months after the first baseline survey. At these later times, the inhabitants of Taibrapa-II village were now located in their alternative residence Taibrapa-I (Fig 1).

We observed that the number of malaria infections fluctuated both spatially and temporally during the 4 month follow-up period, reflecting areas and periods with decreased (2nd survey) and then increased (3rd survey) malaria positivity (Fig 5). Although malaria positivity varied among villages (S1 Table), all cross-sectional surveys were characterized by a predominance of submicroscopic infection. More specifically, while the overall prevalence of microscopic

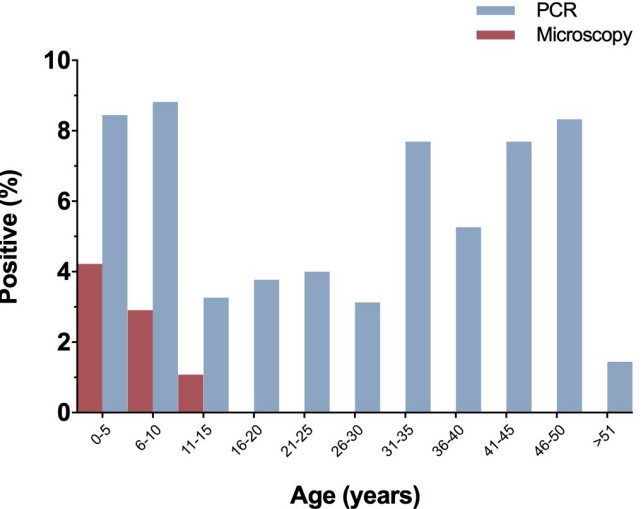

**Fig 4. Prevalence of microscopic and submicroscopic malaria infection stratified by age at baseline survey.**
Submicroscopic infections were determined by PCR-based protocols (PCR) and microscopic infections by
conventional Giemsa-stained thick blood smears (Microscopy).

infections ranged from 0.7 to 2%, PCR ranging from 3.6 to 8.7% (Fig 5), which corresponded
to a four to five-fold increase in the prevalence of malaria.

The prevalence of the three detected *Plasmodium* species also varied during the study
period (Fig 6). While the prevalence of *P. vivax* and *P. malariae* were similar during the first
baseline survey, *P. vivax* infections predominated at the time of the 2nd and 3rd cross-sectional
surveys. A decrease in the prevalence of *P. malariae* infections at the time of the 2nd survey–
that occurred concomitantly with the migration of the inhabitants of Taibrapa to their

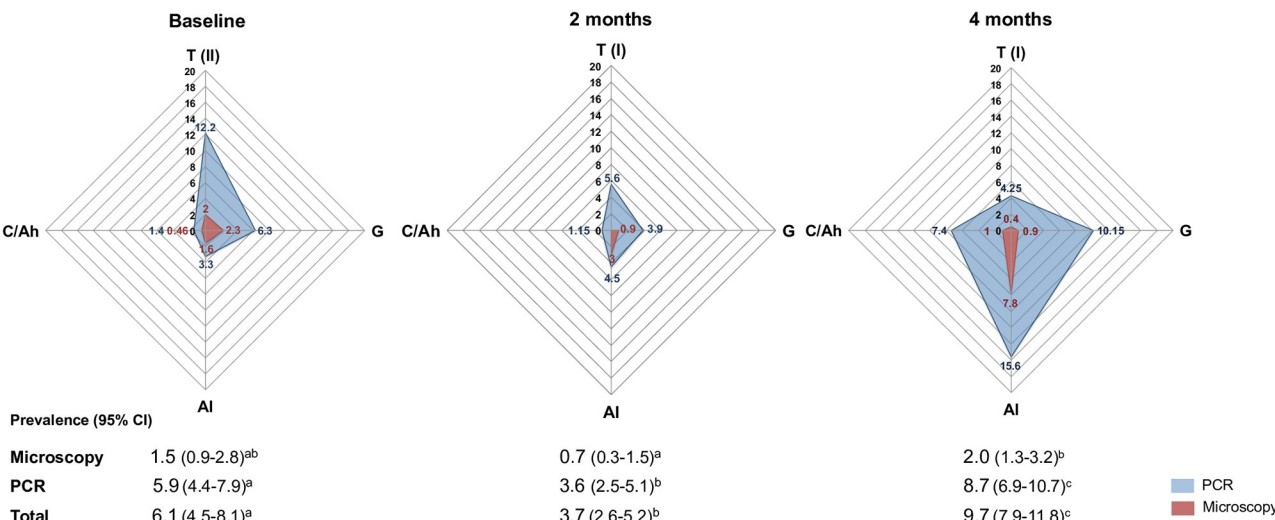

**Fig 5. Radar charts showing the malaria prevalence by village over the 4-month study period.** Three cross-sectional surveys were carried as
described in the legend of Fig 2 (baseline, and then 2 and 4 months later). On each radar chart the data are shown as the percentage of individuals
positive by either conventional microscopy or PCR-based methods. The 4 Yanomami villages are represented as Taibrapa (T-I or T-II, according to the
alternative residences used by the Yanomami during the study), Gasolina (G), Alapusi (Al) and Castanha/Ahima (C/Ah). The overall percentage of
positive individuals and their 95% confidence intervals for each survey are shown underneath their respective radar charts, with different letters (a-c)
indicating statistically significant differences between the surveys (Fisher's exact test, $P < 0.05$).

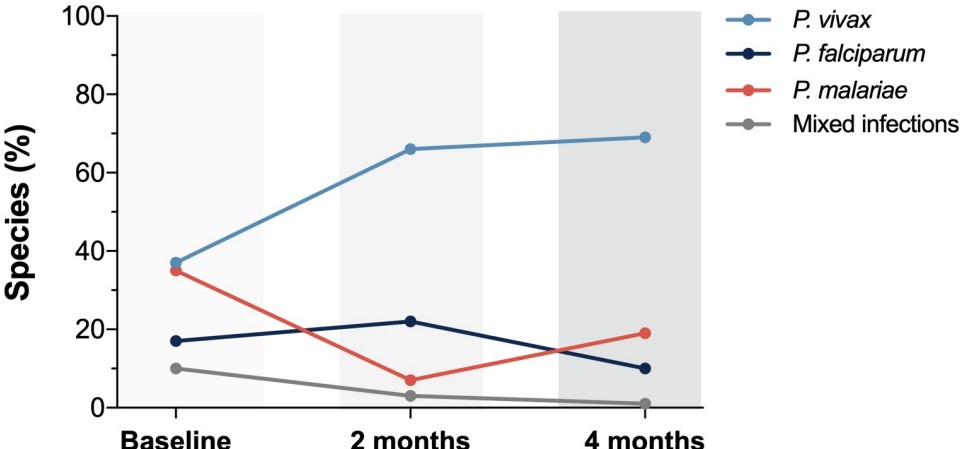

**Fig 6. Comparison of the frequency of species-specific positivity (*P. vivax*, *P. malariae*, *P. falciparum* and mixed infections) during the study period (baseline, 2 and 4 months later).**

alternate residence–resulted in a temporary increased in the prevalence of *P. falciparum* over *P. malariae*. Regardless of the cross-sectional survey, the distributions of the species-specific prevalences of malaria infection by age confirmed that microscopic infections were only present in those under 16 years old, while submicroscopic infections were prevalent in all age groups (Fig 7).

Consecutive blood samples taken from the 859 Yanomami studied allowed us to evaluate within-individual variation in malaria positivity by either microscopy or PCR-based methods. Of the 143 (16.6%) individuals who were positive to malaria at any time during the study (S2 Fig), only 12 (8.4%) had an additional positive sample, most of which were detected using PCR-based protocols (S2A Fig) and were from young children (median age 3 yrs). With regard to single-sample positivity, 35 (24.5%), 22 (15.4%) and 74 (51.7%) individuals were positive to malaria at either the 1[st], 2[nd] or 3[rd] survey, respectively (S2B–S2D Fig). The median age of individuals with a single positive sample ranged from 9 to 11 years old, and there was no statistically significant difference between this age group and those of young children who had consecutive positive samples (3 *vs* 9–11 yrs).

## Discussion

Although malaria is known to cause a significant burden of disease for many populations of indigenous people in Latin America, inequitable access to healthcare leads to sparse and fragmented data on malaria prevalence [7], which are likely to be underestimates due to the use of conventional diagnosis by light microscopy. In order to provide more realistic estimates of malaria prevalence among the Yanomami of the Amazon, we longitudinally surveyed both microscopic and submicroscopic malaria infections in four villages of the Marari community in northern Brazil. Using distinct multi-copy PCR-based protocols–ribosomal and more sensitive non-ribosomal targets [21]–it was possible to demonstrate that in this group of semi-isolated people approximately 75% to 80% of all malaria infections were submicroscopic, with the ratio of submicroscopic to microscopic infections remaining stable overtime. These results are important for disease control in the Brazilian Amazon region as individuals with submicroscopic malaria infection are likely to be asymptomatic and, therefore, remain an untreated and potentially infectious reservoir [24]. The limited previously available data similarly report relatively high prevalences of submicroscopic malaria infection among other communities of

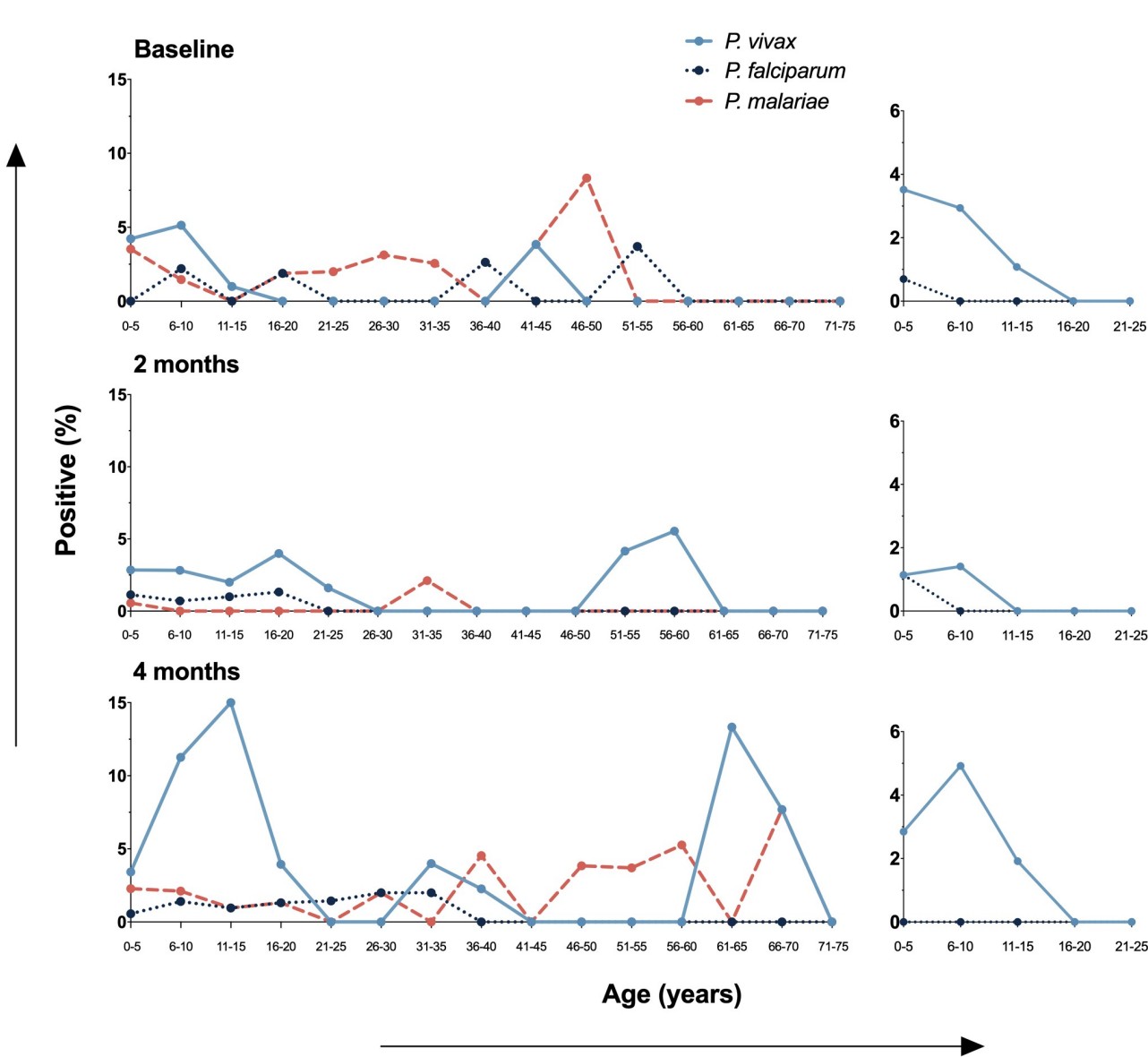

**Fig 7. Species-specific malaria positivity stratified by age during the 4 month study period (baseline, 2 and 4 months later).** Infections caused by *P. vivax*, *P. falciparum* and *P. malariae* were determined by microscopy and/or PCR-based protocols, as described in the methods.

indigenous people in the region close to the border of the Brazilian Amazon, particularly in Venezuela, for which most data are available [11, 25]. Although there is currently no consensus regarding the contribution of submicroscopic malaria infection to the maintenance of malaria transmission, recent findings indicate that such infections may (i) appreciably contribute to the infectious reservoir, (ii) be long-lasting, and (iii) require more sensitive diagnostics in epidemiological settings with lower transmission [26]. Although our findings highlight that

molecular diagnosis is more appropriate for the estimation of the prevalence of malaria in indigenous people in the Amazon, surveillance of malaria infection remains a considerable challenge in hard to reach isolated populations [2].

With regard to gender-related malaria positivity, we detected a similar prevalence of infection in males and females. This result was not completely unexpected as in Yanomami areas, in which transmission predominates outdoors around dwellings, socio-cultural habits and the nomadic behavior of large family groups may result in similar levels of exposure for both sexes [25, 27]. In forest-dwelling people, it has been shown that the cultural activities of the entire family in close proximity of their house during the peak hours of vector activity (e.g., going to the river early in the morning to bathe) may facilitate exposure to infection [28]. In addition, the inhabitants of each Yanomami village reside under a single, temporary, non-partitioned communitarian house (roughly 200 people per "shabono"), built in clearings in the forest, which may also promote equal exposure of males and females to infected mosquito bites. Submicroscopic malaria infection was also equally prevalent among adults and children, but microscopically-detectable parasitemia was identified exclusively in children and adolescents (<16 years old). Due to the limited sensitivity of light microscopy at low parasite densities [29], it is possible to postulate an increased susceptibility of children and adolescents to acute malaria with high levels of parasitemia. Although the current study was not designed to investigate malaria morbidity, our findings are consistent with previous reports showing that severe cases of malaria in Yanomami communities usually occur in children under 10 years old [30]. According, a recent overview of the current state of health of indigenous populations in Latin America confirmed the unfavorable scenario for children and adolescents of persistent high morbidity and mortality rates from infectious and parasitic diseases [31].

In the current study, the prevalence of malaria differed considerably among the Yanomami villages, suggesting high heterogeneity in malaria transmission at both spatial and temporal micro-scales. Local variability in natural breeding habitats of anophelines may account for differences in malaria prevalence between villages and surveys. In a previous study, we described a wide variety of larval microhabitats in the Yanomami area, including river-associated lakes, temporary pools of rainwater and flooded areas, as well as forest streams and rivers [10]. Consequently, different larval habitats with different intrinsic characteristics may result in a high heterogeneity in larval density and *Anopheles* species between indigenous villages and communities [8]. Additionally, as the inhabitants of a given Yanomami village periodically travel as a single large group, exposure to anopheline vectors may also fluctuate according to the time they spend in different residences. During the current study, for example, the inhabitants of Taibrapa traveled between their alternative residences (from Taibrapa-II to I). Coincidentally, we detected a significant decrease in malaria prevalence among the inhabitants of Taibrapa (between the 1st baseline to 2nd cross-sectional survey), possibly reflecting differences in vector exposure between the residences. In addition, infections may be short-lived, and/or the reintroduction of malaria parasites may take time for local transmission to re-establish itself. Overall, our findings are important for guiding and implementing surveillance and control measures, as they indicate that particular locations (i.e., villages) and specific periods of time may be most appropriate for intervention.

Screening for species-specific malaria infections in consecutive cross-sectional surveys confirmed that *P. vivax* is the predominant malaria parasite species circulating in our study area, with regard to both microscopic and submicroscopic infection. *Plasmodium vivax* may be able to persist in the human population since it can relapse after variable periods of dormancy [32] and because attempts at radical cure with primaquine often fail in isolated communities. Indeed, *P. vivax* is typically the most prevalent species in the Amazon area, and the results

presented here are consistent with its historical predominance in indigenous areas, previously detected mainly by routine Giemsa-stained microscopy [7, 27, 30].

Although there was some variation in the prevalence of *P. malariae* between villages and surveys, the molecular protocols we used confirmed that *P. malariae* is present and a frequently detected malaria parasite species in our study area. Previous assessment of malaria infection in the Yanomami indigenous territory also reported the circulation of *P. malariae* in indigenous populations [11, 30, 33], which typically occurs in sympatry with other species of *Plasmodium* [34]. Despite this, only low levels of mixed infection were detected by us, although all but one of them (5 out of 6) included *P. malariae*. The occurrence of human *P. malariae* infection in the forested areas inhabited by the Yanomami is consistent with the possibility that monkeys may be acting as reservoirs, as *P. malariae* appears to be the same species as *P. brasilianum*, which is a malaria parasite commonly found in New World monkeys [35]). Given that *P. malariae* can be associated with renal complications, including acute kidney injury and glomerulopathies (reviewed in [36]), our findings indicate that the effects of this malaria parasite species on indigenous people in the Amazon should be systematically investigated.

Although *P. falciparum* was much less prevalent than *P. malariae*, the results described here demonstrated that this most pathogenic malaria parasite was present in all the villages studied and infected young children. This is important because in the Venezuelan Amazon rainforest, *P. falciparum* has been reported to have a higher prevalence among the Yanomami than other ethnic groups [33]. Unfortunately, as illegal mining and timber operations have moved into the forests of the Yanomami indigenous territory, the periodic introduction of *P. falciparum* has increased [37]. Future studies are needed to evaluate the impact of this increased exposure to *P. falciparum* on the health of Yanomami children.

Finally, our longitudinal study provided an exceptional opportunity to explore changes in malaria parasite positivity over a 4 month follow-up period. Our results showed that only 8% of all positive individuals remained positive in subsequent surveys, with most of such individuals being very young children (median age 3 years old). This observation–together with microscopic malaria infections being more frequent in young children–is consistent with naturally-acquired immunity increasing with age in the Yanomami population as a function of increased exposure to malaria parasites. Clinical immunity to malaria is known to be acquired after repeated parasite exposure, but its rate of acquisition varies widely depending on the intensity of transmission and geographical setting (reviewed in [38]). Our data fit this epidemiological pattern, with malaria parasite densities within infected individuals not remaining constant, but fluctuating and declining with increasing age, and then eventually falling below the detection threshold of molecular assays [39]. Although we were not able to genotype malaria infections over time, it has been suggested that acquired immunity controls transmission mainly by limiting blood-stage parasite densities rather than changing rates of acquisition or clearance of infections [40]. Future studies of the Yanomami could evaluate individual infections based on the patterns of persistence of individual genotypes (i.e., monitor fluctuations in clone-specific malaria parasite density).

The current study had some limitations that should be taken into consideration when interpreting its results. Regarding the best time for consecutive sampling, a major challenge was the remote location of the Yanomami villages within the Amazon forest, whose accessibility is seasonal and dependent on small planes and local boats. Seasons in the Amazon rainforest are not well-defined, and it is divided into the dry season and the wet season, each lasting about six months. Despite this, our cross-sectional surveys covered months of both the dry season (September and November) and the rainy season (January). Consequently, we are confident that the results presented here are representative of temporal variation in malaria prevalence

associated with dry/wet seasons. Overall, our results confirm that molecular methods are a much more sensitive diagnosis tool than traditional light microscopy for the identification of malaria infection among the Yanomami, which are characterized by low-density parasitemias and the circulation of multiple malaria parasite species. Unfortunately, a suitable protocol for nucleic acid amplification (NAATs) in the field is not available yet [41]. Thus, innovative solutions and cost-effective strategies that can accurately identify the real burden of malaria infection in remote and isolated areas are urgently required as part of the global initiative for malaria elimination [42].

## Supporting information

**S1 File. Conditions for the different PCR-based protocols used to amplify genes from *P. vivax*, *P. falciparum* and *P. malariae*.**
(DOCX)

**S1 Fig. Prevalence of malaria according to age group (years) and gender (male and female).** Malaria positivity was defined by PCR-based protocols (PCR) or by conventional light microscopy (Microscopy). The number (*n*) of individuals in each age group is represented in the respective bars.
(TIFF)

**S2 Fig. Species-specific malaria positivity of each positive individual who participated during the cross-sectional surveys.** Each row represents a single individual coded (#) and grouped according to their profile of positive samples in each of the three cross-sectional surveys (columns I, II and III, respectively, of the colored matrix). (A) shows individuals who were positive in at least two of the three cross-sectional surveys (*n* = 12). (B), (C) and (D) show individuals who were positive in only one of the three cross-sectional surveys: either (B) the 1st baseline survey (I; *n* = 35), (C) the 2nd survey (II, *n* = 22) or (D) the 3rd survey (III; *n* = 74). Species-specific PCR positivity is represented using different colors, as indicated in the legend: *P. vivax* (Pv) in light blue; *P. falciparum* (Pf) in dark blue; *P. malariae* (Pm) in orange; and mixed infections by the other different colors. The dark dot (•) inside each square indicates positivity by microscopy as well as PCR, while the asterisks (*) indicate positivity only by microscopy. Individual age, gender and place of dwelling were included in the left part of figure. Each village was coded as according to the legend of Fig 5.
(TIF)

**S1 Table.**
(TIF)

## Acknowledgments

The authors would like to thank the Yanomami people for participating in this study and for their help in many ways during the fieldwork, the Secretaria Especial de Saúde Indígena (SESAI) and DSEI-Y and their employees for logistic support and assistance with the fieldwork, the Program for Technological Development in Tools for Health-PDTIS-FIOCRUZ for use of the Real-Time PCR (RPT09D) facility, and Dr. Luke A. Baton for revising the manuscript.

## Author Contributions

**Conceptualization:** Daniela Rocha Robortella, Cristiana Ferreira Alves de Brito, Tais Nobrega de Sousa, Joseli Oliveira-Ferreira, Luzia Helena Carvalho.

**Data curation:** Daniela Rocha Robortella, Joseli Oliveira-Ferreira, Luzia Helena Carvalho.

**Formal analysis:** Daniela Rocha Robortella, Lara Cotta Amaral, Luiz Felipe Ferreira Guimarães, Michelle Hallais França Dias, Tais Nobrega de Sousa, Joseli Oliveira-Ferreira, Luzia Helena Carvalho.

**Funding acquisition:** Mariza Maia Herzog, Joseli Oliveira-Ferreira, Luzia Helena Carvalho.

**Investigation:** Daniela Rocha Robortella, Anderson Augusto Calvet, Cristiana Ferreira Alves de Brito, Mariza Maia Herzog, Joseli Oliveira-Ferreira, Luzia Helena Carvalho.

**Methodology:** Daniela Rocha Robortella, Anderson Augusto Calvet, Lara Cotta Amaral, Raianna Farhat Fantin, Luiz Felipe Ferreira Guimarães, Michelle Hallais França Dias, Cristiana Ferreira Alves de Brito, Tais Nobrega de Sousa, Joseli Oliveira-Ferreira, Luzia Helena Carvalho.

**Project administration:** Mariza Maia Herzog, Joseli Oliveira-Ferreira, Luzia Helena Carvalho.

**Resources:** Cristiana Ferreira Alves de Brito, Mariza Maia Herzog, Joseli Oliveira-Ferreira, Luzia Helena Carvalho.

**Supervision:** Mariza Maia Herzog, Joseli Oliveira-Ferreira, Luzia Helena Carvalho.

**Validation:** Daniela Rocha Robortella, Lara Cotta Amaral, Raianna Farhat Fantin, Luiz Felipe Ferreira Guimarães, Michelle Hallais França Dias, Cristiana Ferreira Alves de Brito, Tais Nobrega de Sousa, Joseli Oliveira-Ferreira, Luzia Helena Carvalho.

**Visualization:** Daniela Rocha Robortella.

**Writing – original draft:** Daniela Rocha Robortella, Joseli Oliveira-Ferreira, Luzia Helena Carvalho.

**Writing – review & editing:** Daniela Rocha Robortella, Anderson Augusto Calvet, Lara Cotta Amaral, Raianna Farhat Fantin, Luiz Felipe Ferreira Guimarães, Michelle Hallais França Dias, Cristiana Ferreira Alves de Brito, Tais Nobrega de Sousa, Mariza Maia Herzog, Joseli Oliveira-Ferreira, Luzia Helena Carvalho.

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
