## [Decision Letter · Decision Letter 0]

9 Dec 2019

PONE-D-19-31210

Prospective assessment of malaria infection in a semi-isolated Yanomami Amazon indigenous community: heterogeneity of transmission and predominance of submicroscopic infection

PLOS ONE

Dear Dr. Carvalho,

Thank you for submitting your manuscript to PLOS ONE. After careful consideration, we feel that it has merit but does not fully meet PLOS ONE’s publication criteria as it currently stands. Therefore, we invite you to submit a revised version of the manuscript that addresses the points raised during the review process.

The Authors should address to the Reviewer's suggestions. The manuscript should be strongly shortened, avoiding the frequent repetitions, in order to improve the readability. furthermore, the English language should be revised by a native English.

We would appreciate receiving your revised manuscript by Jan 23 2020 11:59PM. To enhance the reproducibility of your results, we recommend that if applicable you deposit your laboratory protocols in protocols.io, where a protocol can be assigned its own identifier (DOI) such that it can be cited independently in the future. For instructions see: http://journals.plos.org/plosone/s/submission-guidelines#loc-laboratory-protocols

We look forward to receiving your revised manuscript.

Kind regards,

Adriana Calderaro

Academic Editor

PLOS ONE

Journal Requirements:

**When submitting your revision, we need you to address these additional requirements:**

**Please ensure that your manuscript meets PLOS ONE's style requirements, including those for file naming. The PLOS ONE style templates can be found at http://www.plosone.org/attachments/PLOSOne_formatting_sample_main_body.pdf and http://www.plosone.org/attachments/PLOSOne_formatting_sample_title_authors_affiliations.pdf**

2.  We note that [Figure 1] in your submission contain [map/satellite] images which may be copyrighted. All PLOS content is published under the Creative Commons Attribution License (CC BY 4.0), which means that the manuscript, images, and Supporting Information files will be freely available online, and any third party is permitted to access, download, copy, distribute, and use these materials in any way, even commercially, with proper attribution. For these reasons, we cannot publish previously copyrighted maps or satellite images created using proprietary data, such as Google software (Google Maps, Street View, and Earth). For more information, see our copyright guidelines: http://journals.plos.org/plosone/s/licenses-and-copyright.

You may seek permission from the original copyright holder of Figure1 to publish the content specifically under the CC BY 4.0 license. 

If you are unable to obtain permission from the original copyright holder to publish these figures under the CC BY 4.0 license or if the copyright holder’s requirements are incompatible with the CC BY 4.0 license, please either i) remove the figure or ii) supply a replacement figure that complies with the CC BY 4.0 license. Please check copyright information on all replacement figures and update the figure caption with source information. If applicable, please specify in the figure caption text when a figure is similar but not identical to the original image and is therefore for illustrative purposes only.

Reviewers' comments:

Reviewer's Responses to Questions

**Comments to the Author**

1. Is the manuscript technically sound, and do the data support the conclusions?

Reviewer #1: Yes

Reviewer #2: Yes

2. Has the statistical analysis been performed appropriately and rigorously? 

Reviewer #1: I Don't Know

Reviewer #2: Yes

3. Have the authors made all data underlying the findings in their manuscript fully available?

Reviewer #1: Yes

Reviewer #2: Yes

4. Is the manuscript presented in an intelligible fashion and written in standard English?

Reviewer #1: No

Reviewer #2: No

5. Review Comments to the Author

Reviewer #1: Malaria infections in remote indigenous population of South American countries such as Brazil are of interest both because of the light they shed on the importance of malaria in such communities and because these remote communities may require special measures to facilitate malaria elimination on this region. This investigation reports results of 3 surveys 2 months apart in Yanomami populations from 4 villages in North West Brazil near the Venezuela border. With over 800 samples collected at each visit, microscopic malaria was restricted to children with adult participants with malaria infection identified only by PCR based techniques. Among infections found, P vivax was most common, with P malariae relatively frequently detected, and P falciparum was rare. There is interesting heterogeneity in parasite positivity between sites, the cause of which this study is not designed to identify.

Major comments

1. This is a very long paper (the pdf generated, including supporting information, had 47 pages). There are also a lot of minor errors in English grammar and usage. This makes it a tiring read. Some general condensing of the text in particular would be helpful, as there is presently quite extensive repetition and (in the discussion) some off-topic speculations. The many different ways the data are presented in the supplementary material, on top of the seven figures in the text, add only marginal value. An expert in English language should be engaged to correct the many minor errors that abound through the manuscript. These are too numerous to record here.

2. The authors misuse the term “malaria cases” which refer to episodes of symptomatic malaria (see Abstract but also text). “Parasitemias” or similar is more appropriate.

3. The lack of infections being repeatedly detected, with no intervention between surveys, is of interest. To my mind the leading explanation is clearance of infections through immune mechanisms (assuming participants did not receive drugs). To support the idea that it is due to infections falling below the threshold of detection would require confirmation e.g. by genotyping infections detected at first and third, but not second visits. These “ultralow density infections” to my mind are controversial- many countries have eliminated malaria in the last decade without being able to find such infections, such that their significance is highly questionable.

4. Methods: while the authors make a significant point regarding the importance of using two different PCR methods to detect infection they do not seem to present any data on the performance of the two. This oversight should be corrected e.g. in a summary table of case detection by different techniques for each species. It does not need to be divided by sample collection or site of origin.

5. Figure 2: the “case reports” presented are not well integrated into the paper and seem of limited relevance to these surveys.

6. Statistics: The number of cases is very small, and no confidence levels are presented around any of the estimates. This makes it difficult to evaluate possible changes over time.

7. Line 411-64: this is much too long a discussion of differences in parasite prevalence between species.

Minor comments

1. The singular of species is species. Specie means money, in coin form. Multiple instances of this.

2. Line 58-60, introduction: surely this figure is incorrect as it omits Venezuela.

3. Line 86-9: example of text that is repeated and or not relevant (e.g.“rich cultural display”). Similarly what is most relevant in lines 100-128?

4. On the other hand, we are told very little about patterns of malaria transmission in these communities. In particular from figure 2 it appears the surveys may have been done at times of lower transmission but this is not described here, or discussed later on. This would have implications for the generalisability of these snapshots.

5. Line 109 masl?

6. Line 232: frequency of microscopic infection, we cannot say it was “acute”

7. Line 318-9: please rewrite this sentence, your point is not clear.

8. Line 331-40: This section on proportions of people infected at more than one time point is important, but it is not clearly presented. In the discussion the authors should discuss the importance of genotyping to determine whether re-infection or persistence are the reason.

9. Line 361-2: “case detection” is not the right terminology. “Detection of infections” would be better.

10. Line 365 onwards: the discussion of gender could be improved. Specifically do men and women undertake similar or different activities within the settlement? AS the authors will know malaria in S E Asia is basically becoming a disease of adult males related often to specific occupations.

11. Line 386: this statement about “influence” does not make sense.

12. 12 Line 4000-1: This decline with translocation suggests infections may be short lived.

Reviewer #2: It is possible to guess what the authors are saying but many sentences are grammatically incorrect. The text would benefit from careful editing by someone fluent in English.

P2 L50: “… the results confirm molecular tools as more appropriate to identify under-registered malaria infection …”: What is under-registered malaria? Asymptomatic Plasmodium infections?

P4 L60: “… malaria incidence began to rise in the Americas, …” when?

P5 Last paragraph of introduction: better to state the research question the authors set out to address than provide a summary of the findings.

Were infected participants diagnosed by PCR offered treatment?

P11 L237 “… the likelihood of having malaria …” does this refer to the prevalence asymptomatic Plasmodium infections? If so, there is a major error running throughout the paper: malaria is defined as evidence of infection with clinical signs. In the absence of clinical signs, the study participants do not have malaria but an asymptomatic Plasmodium infections. Please decide whether you describe clinical malaria or asymptomatic Plasmodium infections? The difference is critical for the comprehension of the study. (If I misunderstood the paper and the authors describe also clinical episodes, they should describe the clinical signs detected.)

P13 “ … PCR-based protocols were able to detect submicroscopic infection in all age groups -- including individuals above of 50 years old -- showing that submicroscopic malaria infections were equally prevalent among adults and children.” If this is true why the drop off in at age 55-60 in Fig 4?

 

Fig 1: map – has three maps inserted on the right. There are no litles what the three maps illustrate/show. Please add legends to each of the insert-maps?

Fig 2: please indicate where the data on malaria cases shown in the chart comes from?

Fig 3: why are only 3 of 5 study villages included?

Fig 4: the x-axis indicates age groups yet the chart/graph is a line/area graph. This does not make a lot of sense if ages are grouped a bar chart is appropriate. If an area graph is used the ages should be continuous variables not age groups.

Fig 5: Great idea to present the data this way! Please explain why is T(II) not included?

Fig 6: lines and areas are confusing. Probably better as bar chart?

Fig 7: the x-axis indicates age groups yet the chart/graph is a line chart. This does not make a lot of sense if ages are grouped a bar chart is appropriate. If an area graph is used the ages should be continuous variables not age groups.

6. PLOS authors have the option to publish the peer review history of their article (what does this mean?). If published, this will include your full peer review and any attached files.

Reviewer #1: No

Reviewer #2: No

---

## [Author Response · Author response to Decision Letter 0]

12 Feb 2020

RESPONSES TO REVIEWERS

REVIEWER #1 – MAJOR COMMENTS: 

(1) This is a very long paper (…). There are also a lot of minor errors in English grammar and usage. (…) Some general condensing of the text in particular would be helpful (…). The many different ways the data are presented in the supplementary material (…). An expert in English language should be engaged to correct the many minor errors that abound through the manuscript. 

AUTHORS: First of all, we thank the reviewer for the careful examination of our manuscript, and consider the results presented here relevant for this field of investigation. As requested, the MS´s text was shortened (mainly methods and discussion) and figures were properly adjusted, with a significant reduction in redundant supplementary figures/Tables. More specifically, we have eliminated Fig. S2; Fig. S3 and Table S1). The English language was revised by a native English speaker (Dr. Luke Baton, https://www.researchgate.net/profile/Luke_Baton/research). We hope that the MS is now suitable for publication. 

 (2): The authors misuse the term “malaria cases” which refer to episodes of symptomatic malaria. “Parasitemias” or similar is more appropriate.

AUTHORS: As requested by the reviewer, in the revised version of the MS we replace “malaria cases” in the text to “malaria infection” or “detection of malaria parasites”, as appropriate. 

However, in some part of text was essential to include the number of malaria cases per 1,000 inhabitants”. To avoid confusion, we included in the methods of the revised version of the MS (please see pages 8/9, lines 180-190) the general consensus definition of “malaria cases” according to the WHO Malaria Policy Advisory Committee as follow “the occurrence of any confirmed malaria infection with or without symptoms.” In the current study, malaria infections were confirmed by either conventional light microscopy (microscopically-confirmed cases) or species-specific PCR-based assays (low-parasite infections or submicroscopic infections).The follow reference was included in the revised version of the MS: WHO Malaria Terminology (https://www.who.int/malaria/publications/atoz/malaria-terminology/en/).

 (3): The lack of infections being repeatedly detected, with no intervention between surveys, is of interest. To my mind the leading explanation is clearance of infections through immune mechanisms (assuming participants did not receive drugs). (…)

AUTHORS: We totally agree with the reviewer that immune response probably contribute for low parasite densities that often fall below the detection threshold of the molecular assays. In fact, immunity is acquired after repeated exposure to malaria, which varies widely depending on the intensity of transmission and geographical settings (revised by Bousema et al, Nat Rev Microbiol,2014; PMID: 25329408). We clarify this topic in the revised MS (Discussion, pages 21/22 ; lines: 495-498, revised version of the MS).

(…) To support the idea that it is due to infections falling below the threshold of detection would require confirmation e.g. by genotyping infections detected at first and third, but not second visits. 

AUTHORS: Unfortunately, the current study was not design to genotype malaria parasites. Furthermore, our research protocol approved by the Ethics Commission for Research on Brazilian Indigenous Populations (CONEP) did not allow us to go beyond the identification of malaria species. Although we were not able to genotype malaria infections over time, it has been suggested that acquired immunity controls transmission mainly by limiting blood-stage parasite densities rather than changing rates of acquisition or clearance of infections (Felger et al., 2012; PMID: 23029082). Future studies in Yanomami area should evaluate the individual infections based on the patterns of persistence of individual genotypes (clone-specific density fluctuations). As requested, this limitation of our study was included in the discussion of revised text (page 22; lines: 501-507).

(…) These “ultralow density infections” to my mind are controversial- many countries have eliminated malaria in the last decade without being able to find such infections, such that their significance is highly questionable.

AUTHORS: Regarding the ultralow density infections, a better understanding of the dynamic of submicroscopic infections in semi-immune populations is necessary to clarify whether there is a need to identify and treat these low-density infections. There are several examples in which microscopy-based detection alone, as part of a robust control programme, resulted in major reductions in the transmission of malaria, but it is unclear whether such reductions could have been achieved sooner if low-density parasitaemias were detected by molecular methods.

 In a recent study published in Nature Communication (Slater et al, Nat Commun.;10(1):1433, 2019; PMID: 30926893), the authors analyzed a series of datasets (15 articles consisting of data from 22 locations in a wide range of transmission intensities), harnessing the increasing quantities of molecular data generated in recent years, to investigate the density, temporal dynamics and infectiousness of low-density malaria infections. The authors demonstrate that although these infections may be less infectious compared with patent infections, submicroscopic infections contribute to the infectious reservoir, may be long lasting and require more sensitive diagnostics to detect them in lower transmission settings. 

Aiming to attend this concern raised by the reviewer, we have addressed this topic in the discussion section (page 18, Discussion, lines: 396-400). 

4. Methods: while the authors make a significant point regarding the importance of using two different PCR methods to detect infection they do not seem to present any data on the performance of the two. This oversight should be corrected e.g. in a summary table of case detection by different techniques for each species. It does not need to be divided by sample collection or site of origin.

AUTHORS: Recently, we described a species-specific non-ribosomal PCR assay with potential to identify low-density P. vivax and P. falciparum infections (Amaral et al., Malar J. 18(1):154, 2019; PMID: 31039781). In that paper, we have compared the performance of non-ribosomal vs. ribosomal PCR assays in clinical and subclinical malaria infections, and demonstrate that the non-ribosomal assay was the most sensitive protocol to detect low-levels of P. vivax/P. falciparum in co-infections. Consequently, the current study was not design to compare PCR assays. Here, we included the non-ribosomal PCR assay to increase the chances to detect possible low-density and co-infections that could be missed by ribosomal PCR assays. 

As requested by the reviewer, we clarified this topic in the Methods section (please see methods, page 10; lines: 216-217 ).

5. Figure 2: the “case reports” presented are not well integrated into the paper and seem of limited relevance to these surveys. 

AUTHORS: We have decided to keep Figure 2 in the MS because it provides relevant information that are now incorporated in the revised version of MS. For that, Fig. 2 was modified and malaria cases were expressed per 1,000 inhabitants. More specifically, data from figure 2 were included in the methods (page 7; lines: 145-147) as well as in the discussion as study/sampling limitations (page 22; lines: 508-518). Thus, we are confident that Fig 2 are now integrated in the MS. 

6. Statistics: The number of cases is very small, and no confidence levels are presented around any of the estimates. This makes it difficult to evaluate possible changes over time. 

AUTHORS: As requested we have included an estimate of confidence intervals to allow a better assessment of the fluctuation of the number of cases over time (please see figure 5, page 15). 

7. Line 411-64: this is much too long a discussion of differences in parasite prevalence between species.

AUTHORS: We have substantially reduced this part of the discussion (about 20 lines shorter, please see track changes version of the MS, all strikethrough lines in red)

REVIEWER #1 MINOR COMMENTS

1. The singular of species is species. Specie means money, in coin form. Multiple instances of this.

AUTHORS: The error was corrected. 

2. Line 58-60, introduction: surely this figure is incorrect as it omits Venezuela.

AUTHORS: As requested by the reviewer, we include an additional sentence showing that in 2018 Venezuela accounted for 50% of all reported cases of malaria in the Americas (World Malaria Report 2019) (please see first paragraph of introduction; page 4, lines 63-66). 

3. Line 86-9: example of text that is repeated and or not relevant (e.g.“rich cultural display”). Similarly what is most relevant in lines 100-128?

AUTHORS: the original Line 86-9 – sentence rewritten; Line 100-128 – the description of the study area was reduced (~ 10 lines shorter, please see track changes version of the MS, all strikethrough lines in red;).

4. On the other hand, we are told very little about patterns of malaria transmission in these communities. (…)

AUTHORS: Few studies so far have reported in detail malaria transmission in the Brazilian Yanomami communities. Despite of that, malaria incidence is unevenly distributed across the Yanomami area, with some areas with reduced malaria receptivity and other localities that are hotspots of intense malaria transmission. The high heterogeneity in anopheline composition, larval habitats and densities between Yanomami communities and villages may explain heterogeneity in local malaria transmission (Sanchez-Ribas et al, 2015,). In the study area, we previously identified high infection rates of Anopheles darlingi (1.5-2%), the main neotropical malaria vector (Sanchez-Ribas et al, 2017; PMID: 29145867). 

Aiming to clarify this topic raised by the reviewer, we have provided more detail about vector transmission in the study area (Methods, page 5, lines: 101-104). Also, this topic was highlighted in the discussion as vector breeding sites may explain heterogeneity of malaria transmission between villages (page 19; lines: 431-439).

 (…) In particular, from figure 2 it appears the surveys may have been done at times of lower transmission but this is not described here, or discussed later on. This would have implications for the generalizability of these snapshots.

AUTHORS: The current study present limitations that should be taken into consideration when interpreting the results. Regarding the best time for consecutive sampling, a major challenge was the remote localization of the Yanomami villages in the Amazon forest whose accessibility is seasonal and dependent on small planes and local boats. Seasons in the Amazon rainforest are not well-defined, and it is divided into the dry season and the wet season, each lasting about six months. Despite of that, our cross-sectional surveys covered months of the dry season (September and November) and rainy season (January). Consequently, we are confident that the results presented here are representative of malaria prevalence in dry/wet seasons. Considering this scenario, we emphasized again the relevance of keep figure 2 in the MS because it shows how variable is malaria transmission in the study area. For example, in January of 2014, malaria infection as detected by conventional microscopy identified 160 cases per 1000 inhabitants; however, in January of next year (2015) only 34 cases/1000 were recorded i.e., a reduction of 5-times. 

We included a paragraph in the discussion section about sampling limitation in the study population (please see discussion, page 22 , lines: 508-518). 

5. Line 109 masl?

AUTHORS: meters above sea level is commonly abbreviated as “masl”. We clarify this topic in the MS. 

6. Line 232: frequency of microscopic infection, we cannot say it was “acute”

AUTHORS: Okay, it was corrected.

7. Line 318-9: please rewrite this sentence, your point is not clear.

AUTHORS: Okay, it was done.

8. Line 331-40: This section on proportions of people infected at more than one time point is important, but it is not clearly presented. In the discussion the authors should discuss the importance of genotyping to determine whether re-infection or persistence are the reason.

AUTHORS: Okay, it was done.

9. Line 361-2: “case detection” is not the right terminology. “Detection of infections” would be better.

AUTHORS: Okay, it was done. However, I would clarify that WHO use “case detection” terminology to classify malaria cases based on a positive microscopy. 

10. Line 365 onwards: the discussion of gender could be improved. Specifically do men and women undertake similar or different activities within the settlement? AS the authors will know malaria in S E Asia is basically becoming a disease of adult males related often to specific occupations.

AUTHORS: We agree with the reviewer that in many endemic areas men have a greater occupational risk of contracting malaria than women if they work in mines, fields or forests at peak biting times. This profile of gender asymmetry in malaria transmission is not restrict to SE Asia but it is also common in the Amazon area (Feitosa Souza, P et al., Plos One, 2019 PMID: 31211772). However, this classical epidemiological model may not apply to semi-nomadic forest dwelling tribes whose activities of men and women during peak biting times may result in equal risks of infection. In these areas, the cultural activities of the entire family outside their house in peak hours of vector activity (e.g., coming to the river early in the morning for bathing or to draw water, fishing, engaging in hunting camps, etc.) may facilitated exposition to infection (Sa et al., 2005; PMID: 15915970). Also, each Yanomami village resides under a single, temporary, non-partitioned communitarian house without walls (“shabono”), built in clearings in the jungle, which may exposure equally male and female to infected mosquitoes-bites. Consequently, it was not surprised that in our study we did not find any statistically significant association between infection and gender. 

As requested, we clarify this topic in the discussion section page 18; lines:405-416).

11. Line 386: this statement about “influence” does not make sense.

AUTHORS: It is possible that the original sentence was not clear enough. Here, the idea is to show that the number of malaria cases fluctuated overtime. In the revised version we clarified this topic.

12. Line 400-1: This decline with translocation suggests infections may be short lived.

AUTHORS: Yes, it is possible that infections are short-lived (we included this possibility in the revised version of the MS).

Thank you very much for all the suggestions to improve the Manuscript.

REVIEWER #2: It is possible to guess what the authors are saying but many sentences are grammatically incorrect. The text would benefit from careful editing by someone fluent in English.

AUTHORS: We thank the reviewer for the careful examination of our manuscript. At this time, we would like to clarify that the English language was revised by a native English speaker (Dr. Luke A. Baton, https://www.researchgate.net/profile/Luke_Baton/research). We hope that the MS is now suitable for publication.

P2 L50: “… the results confirm molecular tools as more appropriate to identify under-registered malaria infection …”: What is under-registered malaria? Asymptomatic Plasmodium infections? 

AUTHORS: We included “under-registered” in the sense that a high proportion of malaria infections were characterized by low parasite densities undetectable by conventional microscopy (i.e., underestimated). To avoid misunderstanding we deleted the word “under-registered”.

P4 L60: “… malaria incidence began to rise in the Americas, …” when?

AUTHORS: Since 2015 malaria incidence began to rise in the Americas, mainly due to increases in Amazon rain forest areas of Venezuela (Bolivarian Republic of), Brazil and Colombia. As requested this information was included in the first paragraph of the introduction (please see page 4, lines: 61-63) 

P5 Last paragraph of introduction: better to state the research question the authors set out to address than provide a summary of the findings.

AUTHORS: As suggested, we stated the research question instead summary of findings.

Were infected participants diagnosed by PCR offered treatment? 

AUTHORS: No, submicroscopic malaria carriers remain untreated in the Brazilian Amazon region. According to the treatment guidelines of the Brazilian Ministry of Health, only microscopically or rapid diagnostic testing (RDT) positive cases are treated for malaria. In the revised version of the MS, this information was included in the discussion (Discussion, page 17; lines: 389-392).

P11 L237 “… the likelihood of having malaria …” does this refer to the prevalence asymptomatic Plasmodium infections? If so, there is a major error running throughout the paper: malaria is defined as evidence of infection with clinical signs. In the absence of clinical signs, the study participants do not have malaria but an asymptomatic Plasmodium infections. Please decide whether you describe clinical malaria or asymptomatic Plasmodium infections? The difference is critical for the comprehension of the study. (If I misunderstood the paper and the authors describe also clinical episodes, they should describe the clinical signs detected.)

AUTHORS: Evaluation of symptoms and pathogenesis was outside the scope of the current study. Consequently, in our study “malaria infection” was defined according to the general consensus proposed by the WHO Malaria Policy Advisory Committee as “the occurrence of any confirmed malaria infection with or without symptoms” (WHO Malaria Terminology; https://www.who.int/malaria/publications/atoz/malaria-terminology/en/). In the present study, malaria infections were confirmed by both conventional light microscopy (microscopically-confirmed cases) and species-specific PCR-based assays (submicroscopic or low-density infections). In this case, we also follow the recommendation that the terms “submicroscopic ” and “asymptomatic” infections were not interchangeably (although often used in the literature generating confusion) as proposed by Meeting report of the WHO Evidence Review Group on Low-Density Malaria Infections 15–16 May 2017 https://www.who.int/malaria/mpac/mpac-oct2017-erg-malaria-low-density-infections-session2.pdf?ua=1.

 Aiming to clarify this relevant topic raised by the reviewer, we included in the Methods the definition of malaria infection used in our study, according to the reference of the WHO malaria terminology publication (please see pages 8/9, lines 180-185). In addition, in the study design we included a sentence to explain that that the terms “submicroscopic ” and “asymptomatic” infections were not interchangeably here (page 9; 185-190 ). Finally, as requested by the reviewer, in the legend of Figure 3 we include the definition of the likelihood of having malaria, i.e., malaria confirmed by conventional microscopy or species-specific PCR-based assays.

P13 “ … PCR-based protocols were able to detect submicroscopic infection in all age groups -- including individuals above of 50 years old -- showing that submicroscopic malaria infections were equally prevalent among adults and children.” If this is true why the drop off in at age 55-60 in Fig 4?

AUTHORS: There were no statistically significant differences between age groups. In the results we have clarified this topic (included statistical data analysis in page 14). 

Fig 1: Map – has three maps inserted on the right. There are no litles what the three maps illustrate/show. Please add legends to each of the insert-maps?

AUTHORS: We added the legends of Figure 1. 

  

Fig 2: please indicate where the data on malaria cases shown in the chart comes from?

AUTHORS: Monthly-time series of microscopically-confirmed malaria cases per 1,000 inhabitants were provided by the National Malaria Surveillance System Registry (SIVEP-Malaria) from the Brazilian Ministry of Health (we included the source of data in the legend of the figure 2, page 8)

Fig 3: why are only 3 of 5 study villages included? 

AUTHORS: The current study involved 4 villages (and not 5): (1) Taibrapa; (2) Gasolina, (3) Alapusi, and (4) Castanha/Ahima (methods, revised MS, page 6; lines: 106-109). However, due to the seminomadic habits of the natives, Taibrapa has alternative residences (I and II). Figure 3 represents the baseline of the study, i.e., the moment when Taibrapa’s inhabitants were temporary located in their area II (named Taibrapa-II). For the odds ratio calculation Castanha/Ahima (lower malaria prevalence) was set as ‘reference category’ with value=1, and then odds ratio for the other three villages (Alapusi, Gasolina and Taibrapa-II) were relative to that reference category. To clarify this topic, we included this information in the legend of figure 3. 

Fig 4: the x-axis indicates age groups yet the chart/graph is a line/area graph. This does not make a lot of sense if ages are grouped a bar chart is appropriate. If an area graph is used the ages should be continuous variables not age groups.

AUTHORS: As requested, Figure 4 was modified to bar chart. 

Fig 5: Great idea to present the data this way! Please explain why is T(II) not included?

AUTHORS: Due to their seminomadic habits, inhabitants of Taibrapa have alternative residences (I and II). In the legend of figure Taibrapa temporary residences were named Taibrapa-I (T-I) or Taibrapa-II (T-II) according to their location during the follow-up period. 

Fig 6: lines and areas are confusing. Probably better as bar chart?

AUTHORS: As requested we have adjusted this figure to make the results easier to read 

Fig 7: the x-axis indicates age groups yet the chart/graph is a line chart. This does not make a lot of sense if ages are grouped a bar chart is appropriate. If an area graph is used the ages should be continuous variables not age groups.

AUTHORS: The bar chart as suggested by the reviewer worked very well in figure 4 but not in this figure. Figure 7 includes so many information that bar chart was not easy to visualize. So, we decided to include “dots” corresponding to each grouped age, and keep connecting lines to show the fluctuation in the prevalence. 

Thank you very much for all the suggestions to improve the Manuscript.

---

## [Decision Letter · Decision Letter 1]

5 Mar 2020

Prospective assessment of malaria infection in a semi-isolated Amazonian indigenous Yanomami community: transmission heterogeneity and predominance of submicroscopic infection

PONE-D-19-31210R1

Dear Dr. Carvalho,

We are pleased to inform you that your manuscript has been judged scientifically suitable for publication and will be formally accepted for publication once it complies with all outstanding technical requirements.

With kind regards,

Adriana Calderaro

Academic Editor

PLOS ONE

Additional Editor Comments (optional):

Reviewers' comments:

Reviewer's Responses to Questions

**Comments to the Author**

1. If the authors have adequately addressed your comments raised in a previous round of review and you feel that this manuscript is now acceptable for publication, you may indicate that here to bypass the “Comments to the Author” section, enter your conflict of interest statement in the “Confidential to Editor” section, and submit your "Accept" recommendation.

Reviewer #1: All comments have been addressed

Reviewer #2: All comments have been addressed

2. Is the manuscript technically sound, and do the data support the conclusions?

Reviewer #1: Yes

Reviewer #2: (No Response)

3. Has the statistical analysis been performed appropriately and rigorously? 

Reviewer #1: Yes

Reviewer #2: (No Response)

4. Have the authors made all data underlying the findings in their manuscript fully available?

Reviewer #1: Yes

Reviewer #2: (No Response)

5. Is the manuscript presented in an intelligible fashion and written in standard English?

Reviewer #1: Yes

Reviewer #2: (No Response)

6. Review Comments to the Author

Reviewer #1: No further comments.I apparently have to wrtie 100 characters as a minimum. The English is now very good.

Reviewer #2: (No Response)

7. PLOS authors have the option to publish the peer review history of their article (what does this mean?). If published, this will include your full peer review and any attached files.

Reviewer #1: No

Reviewer #2: No

---

## [Editor Report · Acceptance letter]

6 Mar 2020

PONE-D-19-31210R1 

Prospective assessment of malaria infection in a semi-isolated Amazonian indigenous Yanomami community: transmission heterogeneity and predominance of submicroscopic infection 

Dear Dr. Carvalho:

I am pleased to inform you that your manuscript has been deemed suitable for publication in PLOS ONE. Congratulations! Your manuscript is now with our production department. 

With kind regards,

on behalf of

MD, PhD, Associate Professor Adriana Calderaro 

Academic Editor

PLOS ONE